# Why are different estimates of the effective reproductive number so different? A case study on COVID-19 in Germany

**Elisabeth K. Brockhaus**[1]*, **Daniel Wolffram**[1,2], **Tanja Stadler**[3,4], **Michael Osthege**[5,6], **Tanmay Mitra**[7,8], **Jonas M. Littek**[1], **Ekaterina Krymova**[9], **Anna J. Klesen**[1], **Jana S. Huisman**[3,10], **Stefan Heyder**[11], **Laura M. Helleckes**[5,6], **Matthias an der Heiden**[12], **Sebastian Funk**[13,14], **Sam Abbott**[13,14], **Johannes Bracher**[1,2]*

**1** Chair of Statistical Methods and Econometrics, Karlsruhe Institute of Technology (KIT), Karlsruhe, Germany, **2** Computational Statistics Group, Heidelberg Institute for Theoretical Studies (HITS), Heidelberg, Germany, **3** Department of Biosystems Science and Engineering, ETH Zurich, Basel, Switzerland, **4** Swiss Institute of Bioinformatics, Lausanne, Switzerland, **5** Institute of Bio- and Geosciences, IBG-1: Biotechnology, Forschungszentrum Jülich GmbH, Jülich, Germany, **6** Institute of Biotechnology, RWTH Aachen University, Aachen, Germany, **7** Department of Systems Immunology and Braunschweig Integrated Centre of Systems Biology (BRICS), Helmholtz Centre for Infection Research, Braunschweig, Germany, **8** Current address: Kennedy Institute of Rheumatology, Nuffield Department of Orthopedics, Rheumatology and Musculoskeletal Sciences, University of Oxford, Oxford, United Kingdom, **9** Swiss Data Science Center, EPF Lausanne and ETH Zurich, Zurich, Switzerland, **10** Physics of Living Systems, Department of Physics, Massachusetts Institute of Technology, Cambridge, Massachusetts, United States of America, **11** Institute of Mathematics, Technische Universität Ilmenau, Ilmenau, Germany, **12** Robert Koch Institute, Berlin, Germany, **13** Department of Infectious Disease Epidemiology, London School of Hygiene & Tropical Medicine, London, United Kingdom, **14** Centre for Mathematical Modelling of Infectious Diseases, London School of Hygiene & Tropical Medicine, London, United Kingdom

* elisabeth.brockhaus@outlook.de (EKB); johannes.bracher@kit.edu (JB)

**Data Availability Statement:** Data and code to reproduce the presented results can be found at https://github.com/ElisabethBrockhaus/Rt_estimate_reconstruction and https://github.com/

## Abstract

The effective reproductive number $R_t$ has taken a central role in the scientific, political, and public discussion during the COVID-19 pandemic, with numerous real-time estimates of this quantity routinely published. Disagreement between estimates can be substantial and may lead to confusion among decision-makers and the general public. In this work, we compare different estimates of the national-level effective reproductive number of COVID-19 in Germany in 2020 and 2021. We consider the agreement between estimates from the same method but published at different time points (within-method agreement) as well as retrospective agreement across eight different approaches (between-method agreement). Concerning the former, estimates from some methods are very stable over time and hardly subject to revisions, while others display considerable fluctuations. To evaluate between-method agreement, we reproduce the estimates generated by different groups using a variety of statistical approaches, standardizing analytical choices to assess how they contribute to the observed disagreement. These analytical choices include the data source, data pre-processing, assumed generation time distribution, statistical tuning parameters, and various delay distributions. We find that in practice, these auxiliary choices in the estimation of $R_t$ may affect results at least as strongly as the selection of the statistical approach. They should thus be communicated transparently along with the estimates.

KITmetricslab/reproductive_numbers. Stable Zenodo releases can be found at https://zenodo. org/record/8343704 and https://zenodo.org/record/ 8343658, respectively.

**Funding:** JB was supported by the Helmholtz Foundation via the project SIMCARD. JB's work was moreover partly funded by the Deutsche Forschungsgemeinschaft (DFG, German Research Foundation) – project number 512483310. SA and SF were supported by The Wellcome Trust (210758/Z/18/Z). The funders had no role in study design, data collection and analysis, decision to publish, or preparation of the manuscript.

**Competing interests:** The authors declare that there are no competing interests.

## Author summary

The effective reproductive number describes how many new infections an individual infected with a given disease causes on average in a population which is subject to a certain degree of immunity and intervention measures. Public health agencies and researchers commonly attempt to keep track of its value over time using various data sources and statistical methods. In this work we compare estimates produced by different research groups in a case study on COVID-19 in Germany. We find pronounced differences between different estimates and shed light on how these are shaped by varying analytical choices. Our results indicate that the employed statistical method has some influence on results, but surrounding analytical choices including epidemiological parameterizations and tuning parameter choices are at least as influential. As estimates are subject to regular updates, we moreover assess how strongly real-time estimates based on different methods were revised retrospectively. While for some methods hardly any retrospective changes occurred, for others there were strong revisions, often incoherent with the uncertainty intervals provided for previous estimates. Our results will be helpful for analysts aiming to set up estimation schemes for the effective reproductive number, and for users confronted with a multitude of potentially disagreeing estimates.

## 1 Introduction

The definition of the effective reproductive number $R_t$ as "the expected number of new infections caused by an infectious individual in a population where some individuals may no longer be susceptible" [1] has become widely known even outside of the scientific community during the COVID-19 pandemic. Values above 1 imply epidemic growth, while values below 1 correspond to a decline. Public health agencies and academic groups from around the world have been publishing $R_t$ values in a daily rhythm since the beginning of the pandemic. In the political debate on the tightening or loosening of intervention measures, these numbers have been routinely cited. Likewise, numerous scientific works on the efficacy of control measures have attempted to link the development of $R_t$ to specific policy choices (e.g., [2–4]).

A major difference between $R_t$ and other epidemiological indicators is that it is not directly observable in practice. While numbers of confirmed cases or occupied hospital beds come with their own problems, they are *data*, i.e., observed values. The effective reproductive number, on the other hand, requires *estimation* unless the complete transmission chain is observed, which is unrealistic in most settings. Estimation is based on statistical models which combine data and epidemiological assumptions, leading to a considerable number of analytical choices to be made. Usually, various defensible options exist, which will influence the results. Estimates produced by different groups of researchers can therefore differ, as is illustrated in Fig 1. The top panel shows estimates of the effective reproductive number of COVID-19 in Germany from January 1, 2021, to June 10, 2021, as published by eight different research teams on July 10, 2021. When taken at face value, these numbers often imply disagreement even on whether $R_t$ was above or below 1. The widths of 95% uncertainty intervals, shown in the bottom panel, vary considerably, and for some pairs of methods, they hardly overlap. In this article, we are concerned with how these discrepancies come about and how they are shaped by different analytical choices.

The pronounced differences between estimates of the effective reproductive number have been pointed out recently by [5]. In an illustration of different estimates of $R_t$ in the United

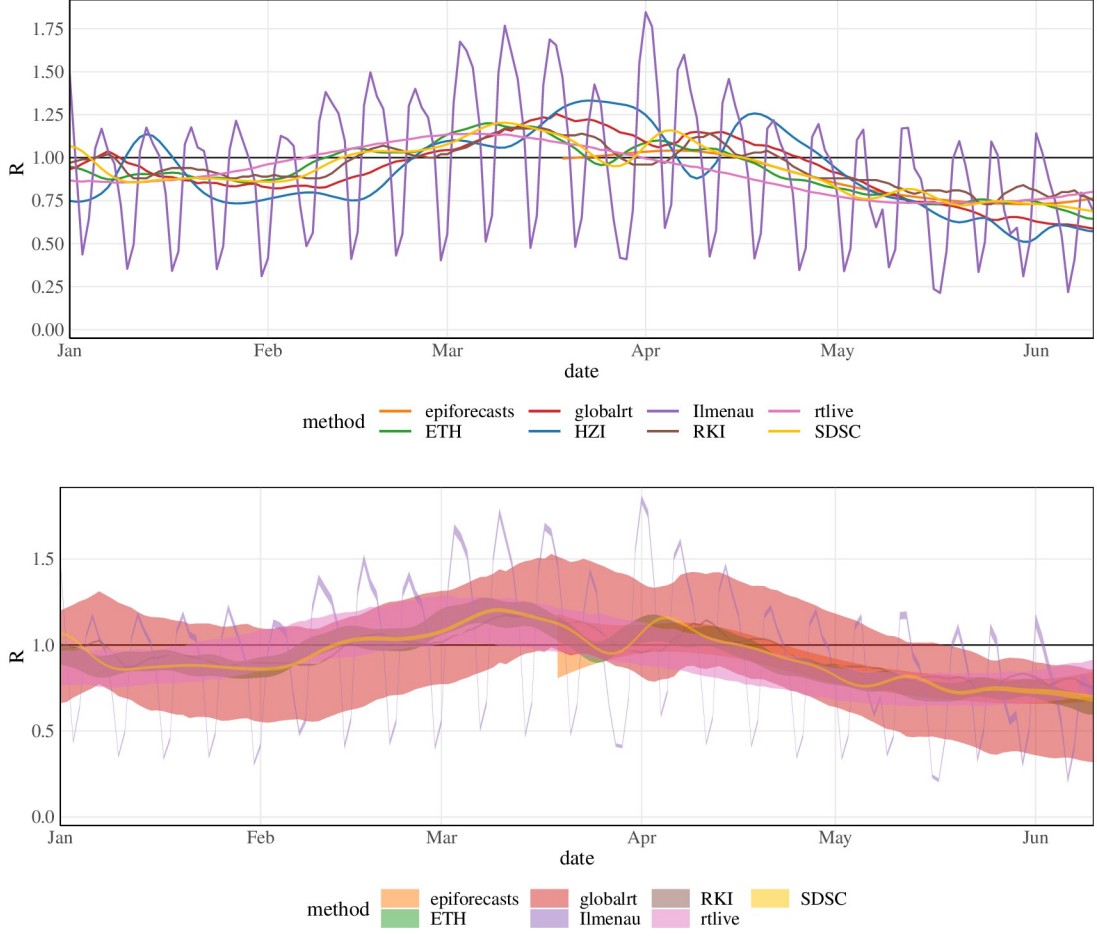

**Fig 1. Overlay of different $R_t$ estimates.** Estimates for the effective reproductive number of COVID-19 in Germany published by eight different research teams on July 10, 2021 (July 11, 2021, for HZI). Top: point estimates (only available for the last 15 weeks for epiforecasts); bottom: 95% uncertainty intervals (not available for HZI).

Kingdom from early October 2020, they found the variability between different estimates to exceed the width of the respective uncertainty intervals. Occasionally, the disagreement between estimates has also spurred confusion in the public debate. For example, on October 27, 2020, Bavarian governor Markus Söder cited an effective reproductive number of 0.57 for his state, which led representatives of the parliamentary opposition to demand a loosening of restrictions [6]. This number, however, differed substantially from the value of 0.9 reported for Bavaria on the same day by Robert Koch Institute ([7]), the German federal public health agency. As clarified subsequently by the Bavarian State Office for Health and Food Safety, Söder had cited an estimate from Helmholtz Centre for Infection Research (HZI, [8]). The statement detailed that the Bavarian authorities monitored estimates from RKI and HZI in parallel, but did not always state the respective source in public communications. The situation is further complicated by the fact that estimates referring to the same day and based on the same method often evolve over time, which has likewise been subject to public debate. As an example, in Fall 2020 it was pointed out that the estimates by RKI were often corrected upwards retrospectively [9].

Given these challenges, a systematic comparative evaluation of $R_t$ estimates is desirable. This, however, is hampered by several conceptual difficulties. Firstly, there is leeway in the technical definition of the effective reproductive number [10], and different approaches may not actually refer to the exact same estimand. Secondly, the effective reproductive number remains a latent quantity even in hindsight. Systematic comparison of estimates and true values is thus only feasible on synthetic data (e.g., [1], [11]). Simulation results, however, will necessarily depend on which model is used to generate data, and it is unclear to what degree they translate to the real world. It has been argued that $R_t$ estimates can be evaluated based on derived short-term forecasts [12]; this, however, is challenging as e.g., errors in the estimated $R_t$ and the assumed generation time distribution may cancel out so that even bad $R_t$ estimates can yield acceptable forecasts. In this work, we take a complementary approach to simulation and forecasting studies by describing discrepancies between real-world $R_t$ estimates and relating them to underlying analytical choices. A somewhat similar approach has previously been taken by [13], who compared $R_t$ estimates based on an SEIR model and the method by [14]. We will analyze $R_t$ estimates for COVID-19 in Germany to study the following aspects:

- *Within-method temporal coherence*: We assess to which degree estimates based on the same method and referring to the same date, but published at different times, vary. In particular, we analyze the agreement of consolidated point estimates with the uncertainty intervals published near-real-time.

- *Between-method agreement of retrospective estimates*: We retrospectively compare estimates across different estimation methods. Reproducing the results published by different groups and harmonizing analytical decisions, we gain insights into how they contribute to the observed discrepancies.

With our analysis we intend to enable readers to critically assess published $R_t$ estimates and make informed decisions when implementing an estimation scheme themselves. The remainder of the paper is structured as follows. Section 2 provides an overview of the different choices an analyst needs to make to estimate $R_t$. Moreover, different estimation approaches applied in real-time to German surveillance data during the COVID-19 pandemic are described. In Section 3 we explore within-method temporal coherence, before turning to the between-method agreement in Section 4. Section 5 concludes with a discussion.

## 2 Analytical choices in the estimation of $R_t$

Estimating $R_t$ requires numerous decisions by the analyst, ranging from the definition of $R_t$ and the statistical approach to epidemiological parameterizations and the choice of the data set (see also [15]). In this section, we review these dimensions and contrast the decisions underlying various routinely published estimates of $R_t$ of COVID-19 in Germany. Table 1 provides an overview of the research groups whose estimates we consider. Most systems were launched throughout the year 2020 (starting with epiforecasts in early March), and by mid-2023 all of them had been retired. Table 2 provides an abridged summary of the model characteristics. For all methods, estimates (and in most cases, analysis codes) were shared under open licences in dedicated repositories, see Section A in S1 Text.

### 2.1 Definition of $R_t$

There are at least two ways of formalizing the concept of the (time-varying) effective reproductive number [1]. The *case reproductive number*, $R_t^{\mathrm{case}}$, quantifies how many new infections individuals who became infected at time $t$ will cause on average. It is thus forward-looking and compares these individuals to the *following* generation of infected. The *instantaneous*

**Table 1. Overview of the groups who regularly published $R_t$ estimates for Germany during the COVID-19 pandemic.** Descriptions of the respective methodology are provided in Section 2.2. Note that web domains provided in footnotes may be discontinued at some point; the links to the repositories provided in S1 Text, Section A, are likely to be more stable.

| Institution | Abbrev. | Reference | Active period |
|---|---|---|---|
| ETH Zurich[1] | ETH | [16] | 2020-05-01–2023-03-23 |
| Robert-Koch Institute[2] | RKI | [17] | 2020-06-04–2023-06-21 |
| Technische Universität Ilmenau[3] | Ilmenau | [18] | 2020-04-22–2023-03-24 |
| Swiss Data Science Center and Institut de | SDSC | [19] | 2020-10-01–2023-06-14 |
| Santé Globale, Université de Genève[4] epiforecasts group / LSHTM[5] | epiforecasts | [20] | 2020-03-02–2022-03-31 |
| Forschungszentrum Jülich[6] | rtlive | [21] | 2020-09-24–2021-07-31 |
| globalrt[7] | globalrt | [22] | 2021-02-15–2023-01-06 |
| Helmholtz Centre for Infection Research[8] | HZI | [23] | 2020-04-29–2023-06-03 |

Links to dashboards:

[1] https://ibz-shiny.ethz.ch/covid-19-re-international/

[2] https://www.rki.de/DE/Content/InfAZ/N/Neuartiges_Coronavirus/Situationsberichte/COVID-19-Trends/COVID-19-Trends.html

[3] https://stochastik-tu-ilmenau.github.io/COVID-19/germany

[4] https://renkulab.shinyapps.io/COVID-19-Epidemic-Forecasting/

[5] https://epiforecasts.io/covid/, previously at https://cmmid.github.io/topics/covid19/global-time-varying-transmission.html

[6] https://rtlive.de

[7] http://www.globalrt.live/

[8] https://gitlab.com/simm/covid19/secir/-/wikis/Report

reproductive number, $R_t^{\mathrm{inst}}$, on the other hand, is backward-looking and compares them to the *previous* generation. Specifically, it is given by the expected number of infections occurring at time $t$, divided by the number of previously infected individuals, each weighted by their relative infectiousness at time $t$. A simple discrete-time display of the recursive relationship between infections $X_t$ occurring on days $t = 1, 2, \ldots$ can help to understand this distinction [24]. For the instantaneous reproductive number, the recursion, also called the *renewal equation*, is given by

$$\mathbb{E}(X_t \mid X_{t-1}, \ldots, X_1) = R_t^{\mathrm{inst}} \times \sum_{i=1}^{t-1} w_i X_{t-i}, \tag{1}$$

where $w_i$ is the probability that the generation time (i.e., the time between primary and secondary infection) equals $i$ time units. Here, the index $t$ in $R_t$ refers to the time of secondary infection. For the case reproductive number the recursion is

$$\mathbb{E}(X_t \mid X_{t-1}, \ldots, X_1) = \sum_{i=1}^{t-1} R_{t-i}^{\mathrm{case}} w_i X_{t-i}, \tag{2}$$

the index $t - i$ in $R_{t-i}$ thus referring to the time of primary infection. We note that $R_{t-i}^{\mathrm{case}}$ can be seen as a convolution of $R_t^{\mathrm{inst}}$ and the generation time distribution [25]. Shifting $R_t^{\mathrm{case}}$ back by the mean generation interval $m$ usually leads to good agreement with $R_t^{\mathrm{inst}}$ (i.e., $R_{t-m}^{\mathrm{case}}$ and $R_t^{\mathrm{inst}}$ can be expected to be similar; [1]). The case reproduction number, however, will change somewhat more gradually; in case of sudden changes discrepancies can thus arise [1, Fig. 2], but these are attenuated if estimation involves smoothing as in most methods considered in our paper.

**Table 2. Methodological characteristics and parameterizations of the compared estimation approaches.** The table follows the structure of Sections 2.1–2.5. The *consensus* model is introduced in Section 4.1 By *conditional distribution of $X_t$* we refer to the distribution of new cases $X_t$ in formulation (1) or (2). The concept of "revision due to smoothing" is discussed in Section 3.3.

**Panel A: Methods based on the Cori method [14] and a consensus parameterization used in Section 4.**

|  | ETH | RKI | Ilmenau | SDSC[1] | consensus |
|---|---|---|---|---|---|
| type of $R_t$ | instantaneous | instantaneous | instantaneous | instantaneous | instantaneous |
| underlying epidemic model | [14] | [14] | [14] | [14] | [14] |
| regularization/prior on $R_t$ | sliding window | sliding window | sliding window | sliding window | sliding window |
| cond. distr. of $X_t$ | Poisson | Poisson | Poisson | Poisson | Poisson |
| inference | Bayesian | max. lik. | max. lik. | Bayesian | Bayesian |
| preprocessing | smooth. + deconv. | nowcast | – | smoothing | – |
| window size | 3 | 7, 4 | 1 | 4 | 7 |
| rev. due to smoothing | yes | no | no | yes | no |
| GT distribution type | gamma | constant | ad hoc | gamma | exponential |
| mean GT (sd) | 4.8 (2.3) | 4.0 | 5.6 (4.2) | 4.8 (2.3) | 4 (4) |
| source of GT | [32] | – | – | [32] | 2 |
| mean IP (sd) | 5.3 (3.2) | 1.0 | 5.0 | – | 0 |
| mean RD (sd) | 5.5 (3.8) | 3.4 | 2.0 | 7.0 | 7 |
| incidence data | RKI, by onset date | RKI, by onset date | RKI, by test date | JHU | RKI, by test date |
| other data inputs | line list | line list | – | – | – |
| programming language | R | R | R | R | R, Python[2] |
| output | posterior mean, 95% HPD interval | point estimate, 95% conf. int. | point est., stand. err., 95% conf. int. | posterior mean, 95% cred. int. | posterior mean, 95% cred. int. |

**Panel B: Other Methods**

|  | epiforecasts | rtlive | globalrt | HZI[1] | |
|---|---|---|---|---|---|
| type of $R_t$ | instantaneous | case | case | instantaneous | |
| underlying epidemic model | [20] | [21] | [22] | [23] | |
| regularization/prior on $R_t$ | Gaussian process | random walk | random walk | sliding window | |
| cond. distr. of $X_t$ | negative binomial | negative binomial | Gaussian[3] | deterministic | |
| inference | Bayesian | Bayesian | Kalman smoother | literature est., least squares | |
| preprocessing | – | – | – | – | |
| window size | – | – | – | 10 & 7 | |
| rev. due to smoothing | yes | yes | yes | no | |
| generation time distr. | gamma | log-normal | exponential | mixt./conv. of exponentials | |
| mean GT (sd) | 3.6 (3.1)[4] | 4.7 (2.9) | 7 (7) | 10.3 (7.6) | |
| source of GT | [33] | [32] | – | – | |
| mean IP (sd) | 5.4 (2.2)[3] | 5.0 | – | 5.2 | |
| mean RD (sd) | 5.9 (14.6)[3] | 7.1 (5.9) | – | 3.7 | |
| incidence data | WHO | RKI, by test date | JHU | RKI, by test date | |
| other data inputs | – | testing volumes | – | mortality | |
| programming language | R | Python | Python | MATLAB, C++ | |

*(Continued)*

**Table 2.** (Continued)

| output | posterior median, 50%, 90% cred. int. | posterior median, mean, stand. dev., 50%, 95% cred. int. | point estimate,65%, 95% conf. int. | point estimate, 100 samples for sensitivity analysis |
|---|---|---|---|---|

[1] Some statements were derived for the present study or retrieved from analysis codes rather than the referenced paper; for details on HZI see Appendix A3.

[2] Python was only used in some data pre-processing steps (adopted from the rtlive code base).

[3] The globalrt model operates on the scale of daily growth rates rather than incidences, but implies a conditional Gaussian distribution for the latter.

[4] The epiforecasts team was the only one to account for uncertainty in the GT, IP and RD distributions; see also Fig 2.

**Abbreviations**: conv. = convolution; cond. distr. = conditional distribution; conf. / cred. int. = confidence / credible interval; deconv. = deconvolution; est. = estimate; GT = generation time; HPD = highest posterior density; IP = incubation period; max. lik. = maximum likelihood; mixt. = mixture; RD = reporting delay; rev. = revision; smooth. = smoothing; stand. dev. / err. = standard deviation / error

## 2.2 Modelling and estimation approaches

Numerous statistical approaches exist to estimate $R_t$ from data. We do not provide a comprehensive review, but focus on methods various research teams have employed in real time to estimate $R_t$ of COVID-19 in Germany (see Table 1). Descriptions are kept concise and we point to the respective references for details.

**Variations and extensions of the Cori method**. Four groups made use of the method by Cori et al [14], but with different parameterizations and data pre-processing. This method uses formulation (1) combined with a Poisson distribution for new cases. Estimation of $R_t$ is then carried out for sliding windows of a width chosen by the analyst. In the widely used R package *EpiEstim* [26] inference is based on a Bayesian approach.

**RKI (Robert Koch Institute, [17, 27])**. First, sampling-based nowcasting is applied in order to impute missing symptom onset dates in incidence data and to correct recent values for reporting delays. Next, the method by Cori et al [14] is applied to each sampled time series, using a fixed generation time and frequentist inference. Uncertainty intervals result from the spread of the $R_t$ estimates across different nowcasting samples. The estimation of uncertainty from the Cori method is not taken into account. The window size is set to either 4 or 7 days. We focus on the latter, which has been used more widely.

**ETH (Swiss Federal Institute of Technology, [16])**. Local polynomial regression (LOESS) is applied to the time series of reported cases to account for weekday effects. The smoothed time series is deconvoluted using various types of delay distributions to reconstruct the time series of infections. $R_t$ is then estimated using the *EpiEstim* package and a window size of 3 days. Uncertainty intervals are obtained by combining the credible intervals and a block bootstrapping approach. The bootstrapping step was only added on January 26, 2021, and led to a widening of intervals (leaving point estimates unaffected). The ETH team published four estimates in parallel (based on confirmed cases as used here, as well as on new hospitalizations, deaths, and test positivity percentages). We focus on the $R_t$ estimates referred to as "sliding window" (the default in the ETH dashboard).

**SDSC (Swiss Data Science Center, [19])**. The case time series is smoothed via a LOESS-based seasonal-trend decomposition prior to estimation using the *EpiEstim* package. The window size is set to 4 days. The proposed extension is focused on the point estimates from the Cori method and does not involve the computation of uncertainty intervals. The provided intervals thus correspond to those returned by the *EpiEstim* package.

**Ilmenau (Technische Universität Ilmenau, [18]).** The effective reproductive number is estimated in a frequentist fashion using Eq (1) and a window size of one day. Wald-type confidence intervals are based on newly derived asymptotic standard errors of the employed estimator.

**epiforecasts (London School of Hygiene and Tropical Medicine, [20]).** The estimation of $R_t$ is based on a Bayesian latent variable approach, implemented in the R package *EpiNow2* [28]. The infection dynamics are modeled as in Eq (1) and linked to the observed case time series via convolutions with the assumed incubation time and reporting delay distributions. The observation model is given by a negative-binomial distribution. A zero-mean Gaussian process with a Matérn kernel is used for the first-order temporal differences of the effective reproductive number with the magnitude and lengthscale estimated jointly with other parameters. Like for ETH, estimates based on hospitalizations and deaths were available, too, but we focus on estimates based on case incidences.

**rtlive (Forschungszentrum Jülich, [21, 29]).** Estimates are based on relationship (2), which is combined with a delay process from infection to detection and a re-scaling of case numbers with inverse testing volumes. Inference is conducted in a Bayesian fashion. Similarly to the epiforecasts approach, a negative binomial observation model is used and $R_t$ is assigned a random walk prior.

**globalrt [22].** This approach exploits a relationship between the epidemic growth rate and the effective reproductive number which holds under the SIR (susceptible-infected-removed) model. The effective reproductive number is assumed to follow a random walk and estimation from observed growth rates is done via a Kalman filter or smoother. We here focus on the smoothing version, which corresponds to a case reproductive number, as this was displayed in the public dashboard. The generation time distribution is assumed to be exponential as in the SIR model.

**HZI (Helmholtz Centre for Infection Research, [23, 30]).** A deterministic SECIR (susceptible—exposed—carrier—infected—recovered) model with time-varying parameters is fitted to cumulative case and death numbers, with certain parameters fixed to or varied around literature estimates. Estimates of $R_t$ are computed from the model parameters, which are estimated for sliding 10-day windows. We use estimates which in addition were smoothed using a 7-day moving average, as shown in the HZI dashboard.

### 2.3 Epidemiological assumptions and parameterization

All described approaches require some parameterization, i.e., specification of epidemiological assumptions. In particular, the distributions of the following durations and delays need to be chosen.

- The *generation time* (GT), i.e., time between primary and secondary infection. The impact of the chosen generation time distribution on $R_t$ estimates is well-studied [31]. The longer the assumed mean generation time, the greater the amplitude of estimates away from 1 (i.e., estimates are increased if $\hat{R}_t > 1$ and decreased if $\hat{R}_t < 1$ for a prolonged period of time). The variance of the GT has a more subtle effect. If $R_t$ is time-constant, $R_t$ estimates are further from 1 the smaller the variance [31]; for time-varying $R_t$, the assumed variance also influences the smoothness of the estimated trajectories.

- The *incubation period* (IP), i.e., time from infection to symptom onset. Changing the mean incubation time shifts $R_t$ estimates in time (as the actual infection events will be assumed to

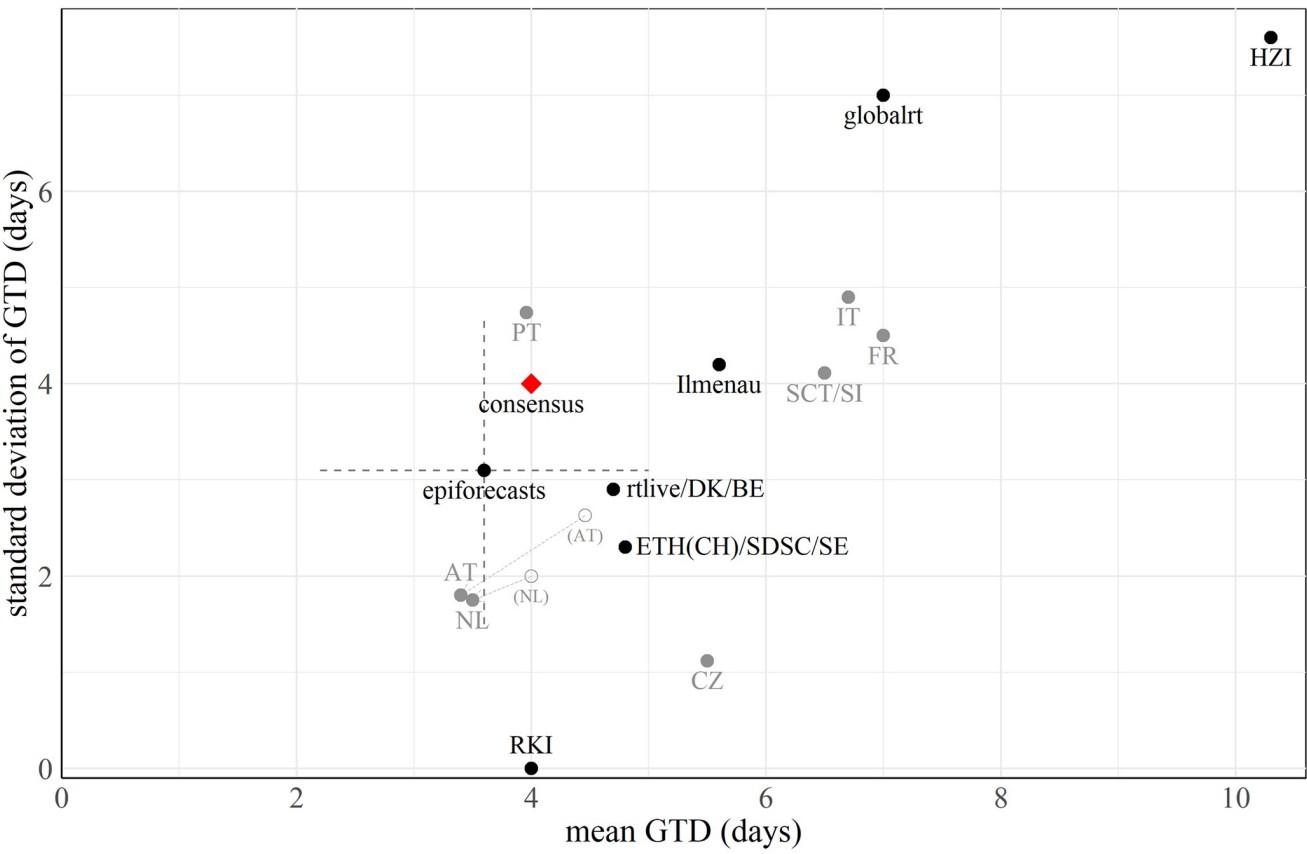

**Fig 2. Scatter plot of mean generation time and corresponding standard deviation used by different research groups.** The red rhombus represents a "consensus value" chosen for further analysis, see Section 4.1. epiforecasts accounted for uncertainty in the generation time distribution by assuming independent normal priors for the mean and standard deviation; we illustrate the respective 95% uncertainty intervals by a cross. For context, we also show values used by public health agencies of other European countries. In the Netherlands (due to the transition to the Omicron variant) and Austria (due to a data update) the parameterization was revised. For details and references see Section B in S1 Text.

precede the respective symptom onsets and reports by a longer or shorter time period). The impact of the variance is not well-studied, but it likely affects the smoothness of estimates.

- The *reporting delay* (RD) between symptom onset and reporting. Like the incubation period, the reporting delay will mainly shift estimates in time.

Table 2 summarizes the distributions used by the different groups. The means and standard deviations of the generation time distributions are moreover displayed in Fig 2. To illustrate that the variability in assumed values is not limited to the German context we added values used by various European public health agencies (see Section B in S1 Text for sources). Note that the values for HZI are not explicitly provided in the manuscript by [23], but have been computed by us based on model parameters reported there (see Section C.1 in S1 Text). The globalrt dashboard allowed users to select a mean generation time between five and ten days; we here use the default setting of seven days.

## 2.4 Methods-specific tuning parameters and prior distributions

The standardized display of analytical choices in Table 2 neglects that in each modeling approach, some additional decisions arise. Bayesian estimation as employed by several teams

requires choosing prior distributions. The HZI approach takes into account numerous epidemiological characteristics other than the generation time, which are informed by literature estimates. The SDSC and ETH approaches involve data smoothing and deconvolution, which require fixing various tuning parameters. These aspects cannot be standardized across methods, and we refrain from analyzing them in detail. Instead, we pragmatically leave them at the values specified by the respective teams wherever needed.

## 2.5 Input data sources

While $R_t$ can also be estimated from death or hospitalization counts [16, 34], we focus on estimates based on COVID-19 case numbers. In Germany and during the considered time period (April 2020—July 2021), such data were regularly released by Robert Koch Institut (RKI, [7]), the World Health Organization (WHO, [35]), and the Center for Systems Science and Engineering at Johns Hopkins University (JHU, [36]). The WHO and JHU data were aggregated by the time cases first appeared in the respective data set. The RKI data were in a line list format containing a reference date called the *Meldedatum* ("reporting date") and for a subset of cases the symptom onset date. The *Meldedatum* denoted when a local health authority digitally registered a case and usually corresponded to the date of the positive test. The Ilmenau, HZI, and rtlive groups aggregated the RKI data by this date. RKI and ETH used the date of symptom onset where available. While RKI completed missing onset dates via multiple imputation, ETH used the reporting date when the symptom onset date was not available and adjusted the reporting delay in the deconvolution accordingly. rtlive additionally used (not publicly available) data on testing volumes, while the HZI model also used mortality data.

Fig 3 shows the different case incidence time series for January through June 2021. The series denoted "RKI, positive test" is aggregated by the date of the positive test using the implementation from rtlive. "RKI, symptom onset" is the time series by symptom onset date as reconstructed by RKI. The series by symptom onset is shifted to the left compared to the others; the WHO data are somewhat shifted to the right, while the JHU and RKI data by test date

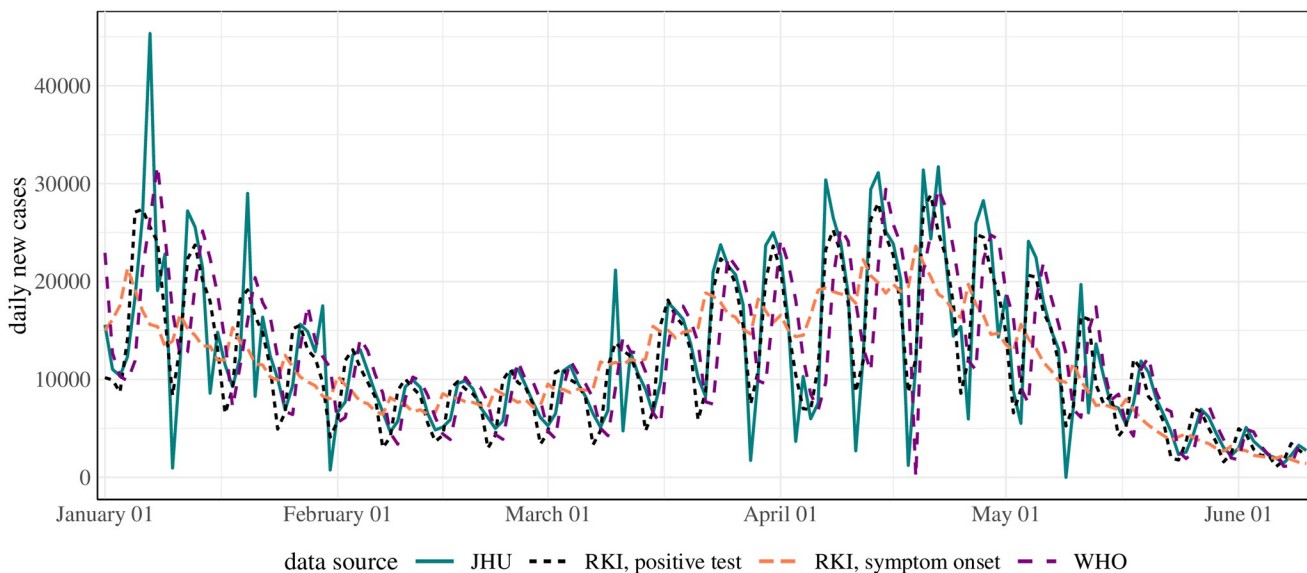

**Fig 3. Case incidence time series used by different research groups.** To enhance visibility we only display the period January through June 2021 (data version: November 23, 2021).

are largely aligned. All series display within-week seasonality, with a smaller amplitude for the RKI data by onset date. The JHU data occasionally display spikes absent in the other series. A last relevant aspect, going beyond Fig 3, is the temporal stability of the data. While the WHO and JHU data were only rarely subject to revisions, the last 3–5 entries in the RKI case data were typically still updated retrospectively.

## 3 Within-model temporal coherence of real-time estimates

We now move to the analysis of $R_t$ estimates based on the methods and parameterizations described before. Estimates were typically updated each day in an automated fashion. Oftentimes these updates also concerned estimates for the past, which were revised in light of new data. Consequently, for each *target date*, i.e., the date to which an estimate refers, a multitude of estimates issued on different *publication dates* are available. This raises the question of *temporal coherence* of estimates. By this, we mean that estimates issued at various times should not differ more than implied by the respective uncertainty intervals. Temporal coherence is a necessary, though not sufficient, prerequisite for reliable estimation. After all, if subsequent estimates from a method are incompatible, agreement with the underlying truth is necessarily limited. Our analyses are based on real-time estimates obtained from the repositories referenced in S1 Text, Section A. These will be compared to *consolidated* estimates issued at a later time point by the same method, using data which can be assumed to have stabilized; see the following subsections for details. We do not explicitly take into account possible modifications of methods during the considered time period; strictly speaking, we thus assess the coherence of estimation systems, which may evolve over time, rather than uniquely defined methods with fixed parameterizations.

### 3.1 Illustrating the evolution of $R_t$ estimates over time

**Visual exploration.**   Fig 4 illustrates how real-time $R_t$ estimates from different methods evolved over time. For each method and a 70-day period, it overlays real-time estimates and a consolidated estimate made six months later (black line) when all data and results can be expected to have stabilized. Where available, 95% uncertainty intervals are shown as shaded areas. We display estimates published on Thursdays where available and published on neighboring days otherwise. Dates of publication are indicated by vertical lines. Note that most teams do not provide estimates up to the publication date, i.e., the $R_t$ trajectories do not reach the vertical line in the respective color. Moreover, some teams (epiforecasts, Ilmenau, SDSC) marked estimates for recent dates as "based on partial data" or "forecast", which we indicate by dashed and dotted lines, respectively. We note that epiforecasts reported 90% rather than 95% uncertainty intervals, along with a standard deviation. As the 90% intervals agreed well with the Gaussian approximation mean ± 1.645× sd, we approximated the 95% intervals as mean ± 1.96× sd.

**Identified patterns.**   Some patterns can be discerned in how and how strongly estimates were revised. While the HZI estimates hardly changed, for RKI and Ilmenau recent values tended to be corrected upwards. The ETH estimates, on the other hand, were mostly corrected downwards for the displayed period. rtlive, epiforecasts and (to a lesser degree) globalrt estimates tended to be corrected upwards when $R_t$ was increasing and downwards in periods when $R_t$ was decreasing. For SDSC, there were some pronounced corrections, but without a clear pattern. Moreover, the approaches differed in the width of the uncertainty intervals. While those from SDSC and Ilmenau were very narrow, those of rtlive and globalrt were so wide that they almost always included the threshold value of 1. For most methods, uncertainty increased for recent dates, leading to funnel-shaped bands. This was particularly prominent

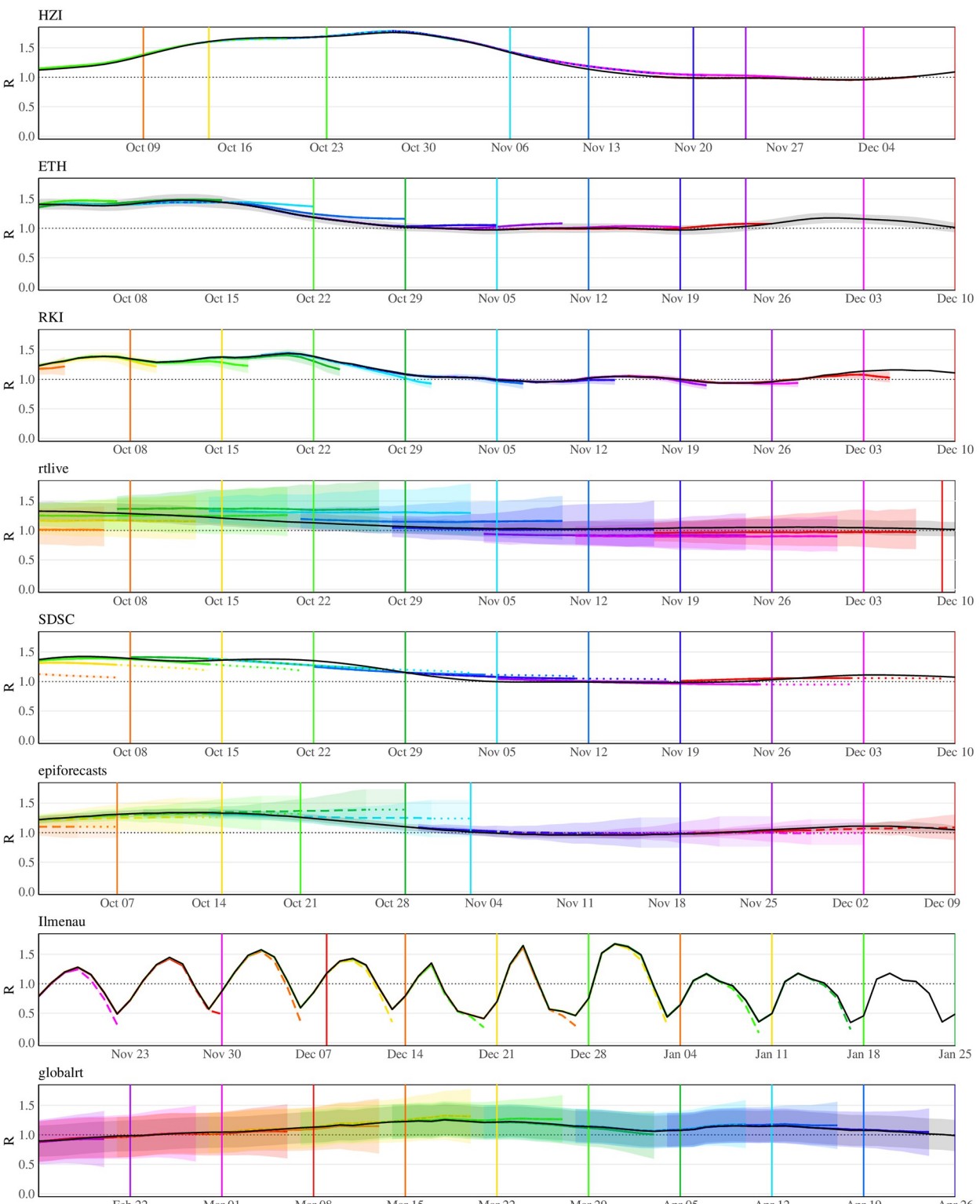

**Fig 4. $R_t$ estimates published between October 1, 2020, and December 10, 2020, and a consolidated estimate published 6 months later (epiforecasts: 15 weeks later).** Note that different time periods are used for Ilmenau and globalrt as these were not operated during the period shown for the other models. The consolidated ETH intervals are wider than those issued in real time due to a revision of methodology. The line type represents the label assigned to the estimate by the respective team: solid: "estimate", dashed: "estimate based on partial data", dotted: "forecast". Shaded areas show 95% uncertainty intervals.

for epiforecasts, whereas the SDSC and ETH intervals were of almost constant width. As mentioned in Section 2.2, the ETH method was revised in early 2021; this change explains why the consolidated intervals are wider than those from Fall 2020. The HZI estimates were published in the form of samples, but it is unclear whether these can be seen as an uncertainty quantification. As estimates were displayed without uncertainty bands on the HZI website, we likewise omit them from panels A and B in Fig 5.

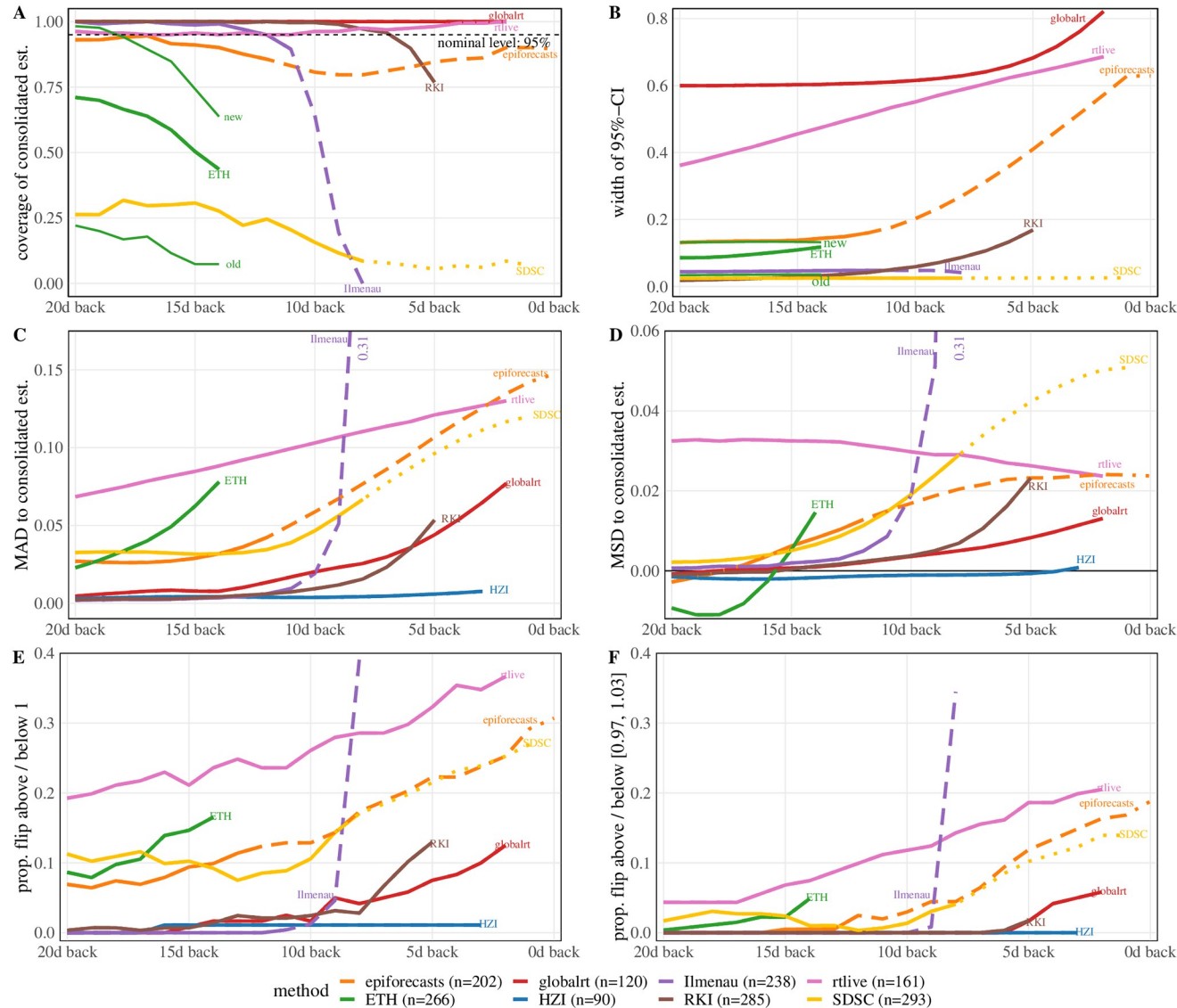

**Fig 5. Temporal coherence of $R_t$ estimates.** Panels: **A** Proportion of 95% uncertainty intervals issued in real-time which contained the consolidated estimate. **B** Mean width of 95%-uncertainty intervals (unavailable for HZI, who only published point estimates). **C** Mean absolute difference of the real-time and consolidated estimates. **D** Same as C, but signed rather than absolute differences. **E** Proportions of cases in which real-time and consolidated point estimates disagree on whether $R_t > 1$. **F** Same as D, but with a tolerance region [0.97, 1.03], i.e., only instances where real-time and consolidated estimates are on different sides of this interval are counted. All indicators are shown as a function of the time between the target date (as stated by the teams) and the publication date. Averages refer to the period October 1, 2020—July 22, 2021 (see Fig F in S1 Text for exact periods during which methods were operated). The consolidated estimate corresponds to the one published 70 days after the respective target date. For ETH two additional lines are included in the top row differentiating between intervals obtained from the old procedure before January 26, 2021 ($n$ = 95), and from the new bootstrap approach afterward ($n$ = 171; see model description in Section 2.2).

### 3.2 Systematic assessment of temporal coherence

**Metrics to assess temporal coherence.** To substantiate these observations, we assess the temporal coherence of estimates quantitatively. Unlike in the illustration in Fig 4 we do not use estimates made at a single later time point as the consolidated ones. Instead, for each target date the consolidated estimate is defined as the estimate generated 70 days later by the same method. This ensures that the time during which the estimates could be revised is the same for all target dates. Based on this definition we computed the following.

- The fraction of instances in which the 95% uncertainty intervals issued in real-time covered the respective consolidated point estimate. If this quantity falls substantially below the nominal level of 0.95, there is an indication that the estimates are temporally incoherent.

- The average width of 95% uncertainty intervals. This serves to contextualize the coverage fractions, which depend on both the degree of revisions and the width of uncertainty intervals.

- The mean absolute difference (MAD) between real-time and consolidated point estimate. This reflects the volatility of real-time estimates relative to the consolidated ones, thus summarizing their temporal stability.

- The mean signed difference (MSD) of real-time and consolidated estimates. This reflects if revisions are systematically in one direction. We orient this such that positive values indicate upwards corrections.

- The fraction of instances in which the point estimate flipped from $\hat{R}_t < 1$ to $\hat{R}_t > 1$ or vice versa. This describes how commonly the qualitative interpretation as epidemic growth or decline changes. Note that this is based purely on the point estimate and ignores the uncertainty intervals.

- The fraction of instances in which the point estimate flipped from $\hat{R}_t < 0.97$ to $\hat{R}_t > 1.03$ or from $\hat{R}_t > 1.03$ to $\hat{R}_t < 0.97$. Instances where an initial estimate close to one crossed this threshold via a minor revision are thus omitted, meaning that we focus on more major revisions.

Fig 5 summarizes the results for estimates published between October 1, 2020, and July 22, 2021. Not all models were operated during the entirety of this period, but we consider it a reasonable overlap (see Table 1 and Fig F in S1 Text on when methods were operated). This period includes two full waves of infections (Fig G in S1 Text) so that effects caused by rising or falling case numbers should largely cancel out. Results are shown as a function of the number of days between the publication date and the target date. E.g., "10d back" means that the estimate refers to the date ten days before the time of estimation. We here stuck to the labeling of estimates by the respective research teams. As they assumed different incubation periods and reporting delays (Table 2), estimates from different methods are not necessarily aligned. Notably, the estimates (and thus curves in Fig 5) by epiforecasts, rtlive, and ETH are shifted to the left relative to the others, as longer incubation periods and reporting delays were assumed. In Fig H in S1 Text we provide a display where curves are aligned to improve comparability. The respective shifts have been determined in a data-driven way, see Section 4.1 and Section D in S1 Text.

**Coherence of uncertainty intervals issued in real time and consolidated point estimate.** Panel A shows the coverage fractions of the 95% uncertainty intervals as defined above. These were in the order of 95% for rtlive and consistently 100% for globalrt. For epiforecasts, coverage was close to nominal less than 4 and more than 14 days back, while there was a

moderate dip in between (this concerned mostly estimates marked as "based on partial data"). RKI and Ilmenau achieved close to complete coverage for dates further back in the past, starting from 9 and 14 days back, respectively. For more recent values, however, coverage dropped. This was particularly pronounced for Ilmenau, with coverage falling to 0% at 8 days back. ETH overall achieved coverage values of 40% to 75% during the period examined in this paper. As can be seen from the additional lines labeled "old" and "new", coverage was considerably higher for estimates published after January 26, 2021, when the computation of intervals was revised (with the explicit goal to account for more sources of uncertainty, see [37]). The coverage of the SDSC (default *EpiEstim*) intervals was around 25% for values labeled as "observed" and dropped to roughly 10% for values labeled "predicted". Panel B shows the average width of the 95% uncertainty intervals. The funnel-shaped character of the confidence intervals of globalrt, rtlive, epiforecasts, and RKI is reflected in the upward shape of the respective curves. As already visible in Fig 4, the uncertainty intervals issued by globalrt and rtlive were considerably wider than those from the other groups. SDSC and Ilmenau issued the most narrow intervals. Prior to the change in methodology in January 2021, the ETH intervals were similarly narrow but became wider afterward.

**Revision of point estimates.**   Panels C and D display the mean absolute and mean signed differences between real-time and consolidated estimates, respectively. For all methods, the mean absolute difference was the largest for recent values. A particularly striking picture is seen for the Ilmenau estimates, where the average correction of estimates 8 days back was 0.31. For some methods the MAD approached zero after a few days (HZI, RKI, Ilmenau, globalrt), indicating that the estimates stabilized. For the remaining models, the average corrections were clearly non-zero even 20 days back, with epiforecasts and SDSC showing a flat pattern from around 12 days back. Panel D shows that for most methods estimates tended to be corrected upwards, especially recent ones. As already visible from Fig 4, this includes RKI and Ilmenau. For ETH the picture is somewhat difficult to interpret, as the sign of the average correction flips at 16 days back. It should be noted that for most models the mean signed differences were much lower than the mean average differences, indicating that corrections in both directions occurred.

**Point estimates flipping above / below 1.**   Panel E shows in which proportion of cases the real-time and consolidated estimates are on opposite sides of the threshold value of 1, i.e., flip between epidemic growth and decline. This is relatively common for recent estimates from Ilmenau, rtlive (more than a third of the cases), and to a lesser degree epiforecasts and SDSC. As can be seen from panel F, these proportions roughly halve if we introduce a tolerance region [0.97, 1.03] and only count instances where the two estimates are on different sides of this region. This implies that many of the flips counted in panel E actually result from minor corrections in cases where $\hat{R}_t \approx 1$. However, some more major revisions remain. We note, though, that even in these cases the uncertainty intervals often contain values on either side of 1, meaning that there is not necessarily a contradiction. For instance, the rtlive intervals very often include values to both sides of 1.

**Sensitivity of results.**   To assess the sensitivity of these results to the definition of the consolidated estimates, we compared estimates published 50 and 70 days after the target date. As Fig I in S1 Text shows, these agree closely. The exact definition of the consolidated estimates is thus not crucial for our results.

## 3.3 Interpretation of observed patterns

We now provide some interpretation of the identified patterns, pointing out possible connections to modeling choices. Retrospective revisions of $R_t$ estimates can stem from two main

mechanisms. These are the revision of input data and information flow due to statistical smoothing assumptions.

**Revisions due to revised impact data.**    Firstly, past incidence values can be revised in the input data, which will lead the same estimation method to produce different results when re-run. The RKI data were subject to such revisions, while the JHU and WHO data were rarely revised. Data revisions were typically upward as delayed reports were added. It seems likely that the strong upward corrections in the Ilmenau estimates stem from this aspect as reporting delays were not accounted for explicitly. We note that data were usually only revised over a few days; afterwards, the Ilmenau estimates thus became quite stable and interval coverage quite high, explaining the characteristic patters in Fig 5. The RKI method included a nowcasting step to account for delays, but the correction seems to have been slightly too weak. The rtlive model accounted for revisions by an empirically determined reporting delay distribution. However, it also relied on testing volume data which was more prone to data revision.

In the Cori and the HZI methods, the length of the estimation window moderates how strongly results can change due to data revisions. The Ilmenau model, which used a one-day window, was strongly affected as estimation hinged purely on the rather unstable last data point. The HZI model, on the other hand, used a ten-day window for estimation and additionally smoothed the consolidated estimates via a trailing seven-day moving average. The consolidated estimates were thus based on a 16-day window (with some weighting). As the revisions of the RKI data only concerned a small part of this window, the resulting revisions of estimates were negligible. We illustrate this in Fig D in S1 Text, which shows that without the additional smoothing step slightly more pronounced revisions of estimates occurred.

**Revisions arising from smoothing assumptions.**    Another reason why estimates may change is smoothing during the estimation process. This can enter either via data pre-processing (ETH, SDSC) or model assumptions on the $R_t$ trajectory (Gaussian process assumption in epiforecasts, random walk in rtlive and globalrt). Via smoothing, a new data point can influence how the model treats previous data, and thus impact the results for preceding target dates. We note that smoothing is a planned feature of the approaches in question. Indeed, estimates up to the day of estimation as available from epiforecasts would not be feasible without a generative assumption implying some smoothness. The trade-off is that near-real-time estimates are increasingly extrapolations of the previous $R_t$ trajectory, and likely to change once more data become available. This explains why estimates from epiforecasts, globalrt, and rtlive were often corrected upwards when $R_t$ was on the rise and downwards when it was on the fall. For methods based on trailing estimation windows (RKI, Ilmenau, HZI) revisions cannot arise from this aspect, even though window sizes larger than one day also imply some smoothing.

**Width of uncertainty intervals.**    Lastly, how well uncertainty intervals cover consolidated estimates depends on how wide the former are. By issuing wide intervals, globalrt and rtlive achieved high coverage despite substantial revisions. While we defer a general discussion of the interval widths to Section 4, we provide some remarks on the widening of intervals for target dates close to the publication date. This funnel-like pattern was particularly pronounced for epiforecasts, globalrt, and rtlive. These methods provided estimates closest up to the publication date, which as mentioned before, got less and less constrained by data. In the Bayesian framework, this translated naturally to wider uncertainty intervals. In the case of rtlive, this was reinforced by hard-coded assumptions on the variability of the random walk. In the RKI approach, the uncertainty from the nowcasting step was forwarded to the $R_t$ estimation, leading to similarly expanding intervals. For both epiforecasts and RKI, this widening was not quite pronounced enough, however, and interval coverage fell below the minimum desired level of 95%. The Ilmenau, ETH, and SDSC (default *EpiEstim*) approaches showed little to no widening of intervals. The likely reason is that the uncertainty about the recent data points was

not forwarded to the $R_t$ estimation from earlier preprocessing steps (see the discussion section of [18] on additional sources of uncertainty). In all three cases, this led to a drop in 95% interval coverage below 50%.

## 4 Between-method agreement of retrospective estimates

We now turn to the agreement across estimates by different research groups, which as shown in Fig 1 can differ substantially. Our approach is to standardize analytical choices in order to assess their contribution to the overall disagreement. This is inspired by the *vibration of effects* framework [38], which for observational studies serves to assess the sensitivity of effect estimates to aspects like model choice and measurement errors. While e.g., the impact of the assumed generation time distribution on estimates is well-understood at a theoretical level (see Section 2.3), we aim to answer an empirical question: What differences arise in practice when different researchers independently take the necessary analytical decisions?

### 4.1 Sequential standardization and individual variation of analytical choices

**The consensus setting.**    As visible from Table 2, the available $R_t$ estimates are not only the results of different statistical methods but also of different parameterizations and input data. Isolating the contributions of these aspects requires standardizing the remaining dimensions as far as possible. In what follows we describe a "consensus setting" which we implement for each of the represented methods (see also Panel A of Table 2, last column).

- *Incidence data*: We use RKI data, which are the most common choice among teams. We use data by test date as aggregated by rtlive for all methods requiring a simple time series. Models making use of information on symptom onset dates (RKI, ETH), test positivity percentages (rtlive) or mortality (HZI) can keep using these as we consider this an integral part of their method.

- *Epidemic model*: We employ the Cori method [14], a common building block in the considered approaches, in its basic form without any pre-processing steps.

- *Window size*: When applying the Cori method we use a window size of 7 days. This is a common choice as it reduces fluctuations arising from within-week reporting patterns.

- *Generation time distribution*: We assume an exponential distribution with rate 1/4, i.e., mean and standard deviation equal to 4 days. While an exponential distribution may not be the most common choice to match the epidemiology of COVID-19, this enables us to include the globalrt model, which can only accommodate an exponential GTD.

- *Incubation period and reporting delay*: These aspects are challenging to standardize across methods, as variation in delays is an integral part of some methods (e.g., epiforecasts) but incompatible with others (e.g., Ilmenau, SDSC). Temporal misalignment resulting from these aspects is therefore handled pragmatically by shifting estimates in time. As the consensus setting, we assume that the reporting delay and incubation period sum up to seven days and shift estimates accordingly.

- *Definition of $R_t$*: By using the Cori method, we estimate instantaneous reproductive numbers. Based on the notion that $R_{t-i}^{\text{inst}}$ lags behind $R_{t-i}^{\text{case}}$ by one mean generation time (see Section 2.1), we again resort to shifting estimates of case reproductive numbers in time to align them.

In practice, it proved challenging to determine exactly how estimates needed to be shifted to account for differing assumptions on incubation periods, reporting delays, and type of $R_t$. Following [39], we therefore adjust temporal shifts for each method in a data-driven way by minimizing the mean absolute difference to the consensus estimates (see Section D in S1 Text for details and additional analyses based on reported delay distributions).

We moreover note that while all other approaches can be reproduced with standardized settings, some compromises are necessary for the HZI model. The input data already correspond to the consensus choice and the various delay distributions are handled by shifting estimates (as for all other models). The generation time distribution, however, cannot be set directly to the consensus setting, as it is not an independent parameter in the HZI model. Instead, it arises from the interplay of numerous other parameters. We therefore opt to transform the published estimates using a relationship linking the generation time distribution and $R_t$ estimates from [31] (see also Section C.2 in S1 Text).

**Sequential standardization.** It is not practically feasible to assess all combinations of standardizing or not standardizing the different analytical choices. We therefore vary them in two specific fashions. In the first procedure, we start from the original settings used by different teams. Then, in the above order, we standardize all analytical choices apart from the statistical estimation approach (including possible pre-processing steps). We refer to this as *sequential standardization*. The order of steps is motivated as follows. It seems natural to handle the data source first (Step 1), establishing a common basis for the remaining aspects. As the choice of window size has a strong impact, we handle it second (Step 2), thus avoiding that it obscures the picture in the following steps. As the temporal shift is handled in a data-driven way, it needs to be handled last (Step 4), leaving the generation time distribution to be handled in Step 3.

**Individual variation.** The second procedure starts from the consensus model (i.e., a simple application of the Cori approach) and subsequently varies the different analytical choices one by one. We refer to this as *individual variation*. An advantage of individual variation is that it does not require specifying an order in which the various dimensions are aligned. The sequential approach, on the other hand, helps to illustrate the compounding of the various effects.

**Retrospective approach.** As some of the considered approaches are computationally costly (in particular the Bayesian hierarchical models by epiforecasts and rtlive) it is not feasible to re-run the estimations under different parameterizations for all considered estimation dates. We therefore refrain from mimicking a real-time setting and assess between-method agreement retrospectively for a single estimation date. Specifically, we consider estimates for the period April 1, 2020, until June 10, 2021, based on data as available on July 10, 2021.

## 4.2 Results for point estimates

**Sequential standardization.** Fig 6 shows how the agreement between methods improves step by step in the sequential standardization of analytical choices. The visual impression of closer and closer alignment from the left column is confirmed by the matrices of mean absolute differences in the right column. These range from 0.03 (epiforecasts vs. ETH vs. SDSC) up to 0.32 (Ilmenau vs. HZI) for the original versions of the estimates, with an average pairwise value of 0.15. This is a substantial difference given that the estimates are mainly between 0.75 and 1.25. Once all analytical choices other than the estimation method and data pre-processing are aligned, mean absolute differences range from 0.01 (ETH vs SDSC) to 0.07 (epiforecasts vs rtlive). Particularly strong improvements result from standardizing the window size where applicable and the generation time distribution. Aligning the window size removes the

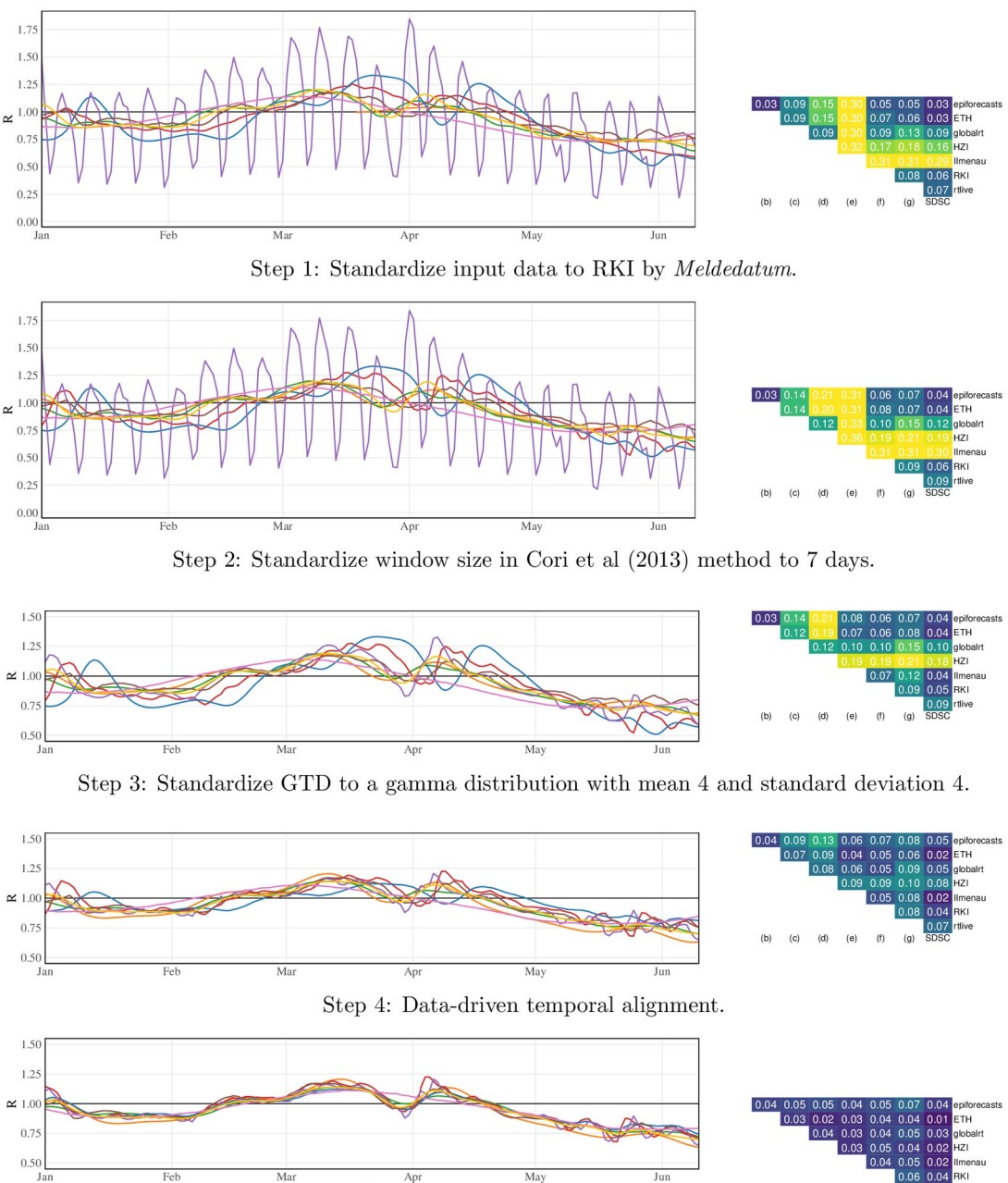

**Fig 6. Step-by-step alignment of analytical choices to the consensus specifications.** The left column shows the resulting $R_t$ estimates for a subset of the considered time period. The right column shows the mean absolute differences between point estimates obtained from the different approaches. In the bottom panel all considered aspects other than the estimation method (incl. data pre-processing) are aligned. Note that the two top rows we use wider y-axis limits to accommodate the Ilmenau estimates.

periodic fluctuations in the Ilmenau estimates, which are based on a window of just one day. Standardizing the generation time distribution has a strong impact on the HZI estimates, which use a long mean generation time of 10.3 days.

As can be seen from the improvement between Steps 3 and 4, temporal shifting of estimates is necessary to achieve good alignment. This shift, which is determined in a data-driven way, accounts for differences arising from the assumed incubation periods and reporting delays and the choice between case and instantaneous reproductive numbers. In almost all cases the shifts agree well with what would be expected based on the respective model descriptions, see Table A in S1 Text. An alternative display where shifts are determined based on these descriptions is available in Fig H in S1 Text. A version of Fig 6 with mean relative rather than absolute differences is available in Fig J in S1 Text, but looks similar.

**Individual variation.** Results for the individual variation approach are shown in Fig 7. Here, we also vary the data pre-processing step separately; this corresponds to nowcasting for RKI, smoothing for SDSC, and a combination of nowcasting, smoothing, and deconvolution for ETH. Pre-processing as well as the choice of data source impact the smoothness of the estimates, but in terms of mean absolute deviations play a limited role. The window size and generation time distribution have a stronger impact on the results. The resulting mean absolute differences are in fact more pronounced than when varying the estimation approach (bottom panel). As implied by theory, the estimates are fanned out away from $R_t = 1$ when longer mean generation times are used. In particular, the HZI choice with a mean generation time of 10.3 days stands out. Concerning the window length in the Cori approach, choices that are not multiples of 7 lead to periodically fluctuating estimates. We note, however, that the ETH and SDSC teams, who use widths of 3 and 4 days, employ data pre-processing steps to suppress this behavior.

## 4.3 Some remarks on uncertainty intervals

**Visual comparison.** An analog display of the bottom panels of Figs 6 and 7 showing 95% uncertainty intervals can be found in Fig 8. Here, all analytical choices have been standardized apart from the estimation method and data pre-processing. While similarly to the point forecasts, the intervals are more aligned in terms of their temporal course, considerable differences in their widths remain. Rather narrow intervals are produced by the Ilmenau, SDSC, RKI, and ETH approaches (based on the updated version of the method). The intervals obtained from the epiforecasts, globalrt, and rtlive methods are wider. This divide coincides with variations of the Cori method on the one hand, and more complex hierarchical approaches on the other.

**Including overdispersion in the Cori method.** One particularity of the Cori approach [14] compared to the three others is that it combines Eq (1) with a conditional Poisson distribution of observed case counts. The epiforecasts and rtlive approaches assume a negative binomial distribution and globalrt implicitly assumes a Gaussian distribution. These distributions, unlike the Poisson distribution, have a free parameter steering the degree of dispersion. It is known that in generalized regression, assuming a Poisson distribution can lead to an underestimation of standard errors when the data are actually over-dispersed [40]. To assess whether this aspect plays a role in the observed patterns we re-ran the Cori method swapping the Poisson for a negative binomial distribution. As can be seen from Fig 9, this results in considerably wider uncertainty intervals, comparable to those from globalrt.

We note that the negative binomial version of the Cori method needed to be newly implemented. For technical ease and to avoid having to specify prior distributions we performed frequentist estimation via the function `glm.nb` from the R package `MASS`. The overdispersion parameter of the negative binomial distribution was estimated jointly with $R_t$ (under the

Varying the input data source.

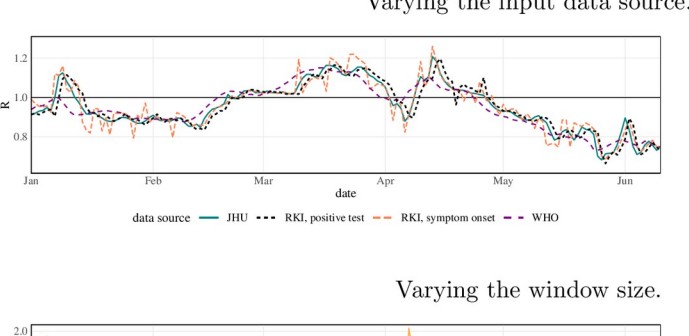
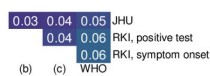

Varying the window size.

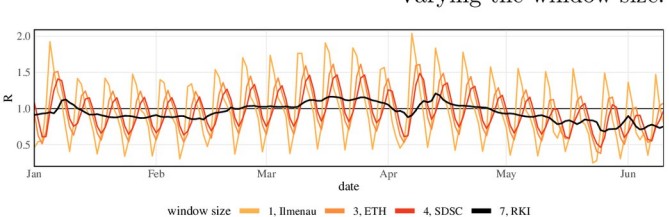
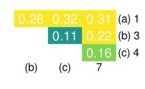

Varying the generation time distribution.

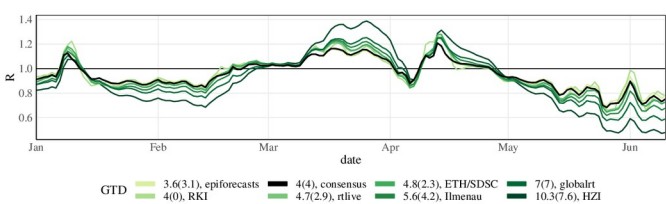
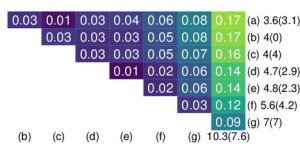

Varying the data pre-processing step.

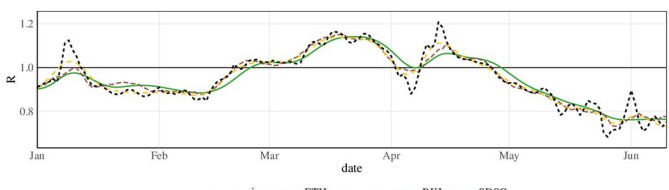
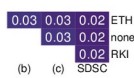

Varying the estimation approach incl. data pre-processing step.

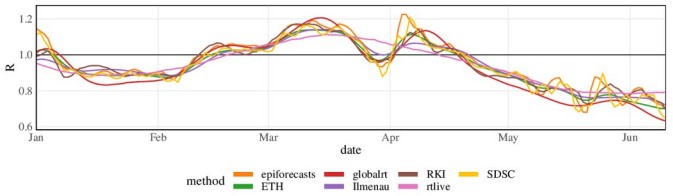
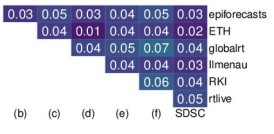

**Fig 7. Individual variation of analytical choices in the consensus model.** Left column: $R_t$ estimates for a subset of the considered time period. Right column: mean absolute differences between point estimates. The values over which the respective quantities are varied correspond to those chosen by the different teams. For the generation time distribution, we adopt the notation mean (standard deviation). Note that the different panels use different y-axis limits. The bottom panel is identical to the one of Fig 6.

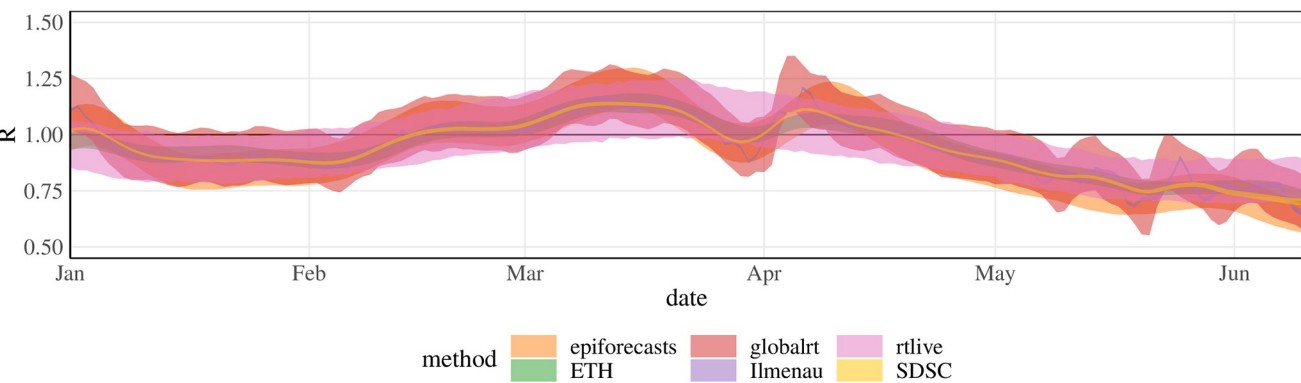

**Fig 8. Comparison of uncertainty intervals after standardization of analytical choices.** The figure shows 95% uncertainty intervals corresponding to Fig 6, Step 4.

assumption of a constant value over the 7-day estimation window); see Section F in S1 Text for details. As an analog implementation of the Poisson version yielded almost identical results to *EpiEstim* we consider the use of a frequentist rather than Bayesian implementation unproblematic.

**Potential role of generative models for $R_t$.** Another potentially relevant difference between the Cori approach and the three others involves the assumptions on the process governing $R_t$. While the Cori method assumes $R_t$ to be constant on a certain time window, the others assume truly generative models, specifically random walks or a Gaussian process. Both assumptions serve to stabilize estimates. However, it is difficult to assess their impact, as replacing them would be a fundamental change to the respective models.

## 5 Discussion

In this paper, we assessed temporal coherence and between-method agreement of $R_t$ estimates for COVID-19 in Germany. We found that for most considered methods, the real-time estimates for dates close to the publication date were subject to substantial revisions. In many cases, these were more pronounced than implied by the accompanying uncertainty intervals. Some methods were able to avoid temporal incoherence but at the cost of wide uncertainty

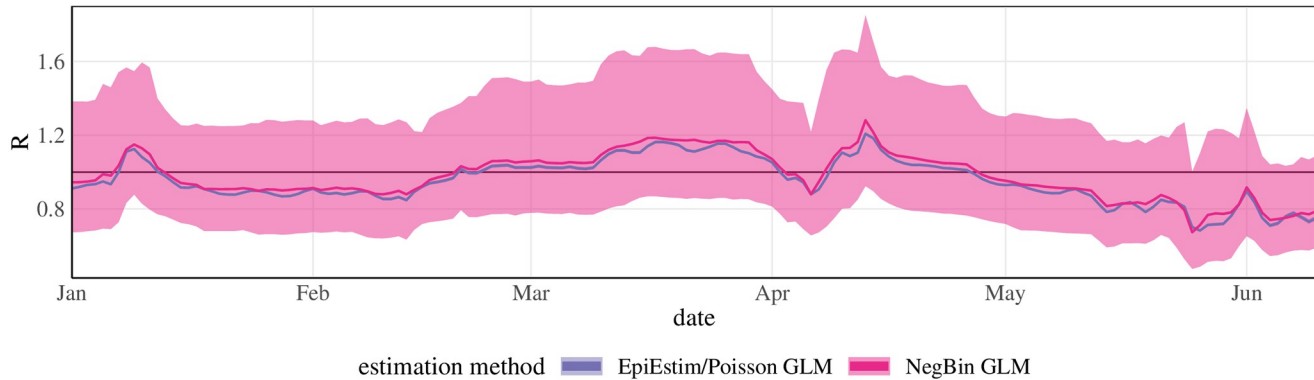

**Fig 9. Comparison of 95% uncertainty intervals of the Cori method (consensus settings) with a Poisson (dark) and negative binomial distribution (light).** The uncertainty intervals under the Poisson distribution are hardly discernible from the line representing the point estimate.

intervals. In our retrospective assessment of the between-method agreement, we found that while the choice of estimation method led to some discrepancies, surrounding analytical choices, e.g., on the generation time distribution, were at least as influential. In terms of the uncertainty intervals of retrospective estimates we found a certain divide between approaches based on the method by Cori et al [14] and more sophisticated Bayesian approaches, which yielded wider uncertainty intervals. As discussed in Section 4.3 the former may tend to underestimate the overall estimation uncertainty, which arises from the compounding of various factors.

Our assessment of temporal self-coherence highlights the importance of continuously tracking the real-time behavior of $R_t$ estimates. If these are overly fluctuating or subject to systematic corrections, this may lead to a loss in user trust. However, the stability of estimates is not the only relevant goal, and there is a trade-off with the timeliness of estimates. $R_t$ estimates are quickly outdated, and results for recent days are the most relevant for public health purposes. These are unavoidably subject to increased uncertainty. This needs to be acknowledged by users, and uncertainty needs to be quantified and communicated appropriately. We believe that analyses of temporal coherence as presented in our work can be a useful tool to this end.

In our between-methods comparison of estimates, we found that in particular the assumed generation time distribution and the choice of estimation window sizes drove differences between estimates published by different research teams. These decisions and their potential impact should thus be communicated transparently. The approach taken by globalrt, where users can vary the mean generation time, is promising, though some contextualization on which values are well-supported by the state of research may be helpful. Temporal shifts arising from different assumptions on incubation periods and reporting delays proved relevant, too, as they shift $R_t$ estimates in time. This is of particular importance when linking the latter to intervention measures. The respective delay distributions should thus be chosen with care. We expect that careful deconvolution of incidence data may yield a clearer picture than simpler shifting or smoothing approaches especially when $R_t$ changes abruptly; this, however, would need to be assessed in simulation studies where the true $R_t$ values are known.

Given the important role of epidemiological parameterizations, we recommend to assess their plausibility carefully when selecting among different sets of $R_t$ estimates. The parameterizations should be backed by recent and solid evidence from the literature. In practice, many $R_t$ estimates for COVID-19 continued to be based on rather uncertain evidence from the early phases of the pandemic, without updates in the light of new evidence or virus properties. The ETH team retrospectively updated their generation time distribution to a new estimate for the Omicron variant [41] in October 2022; a similar change was made e.g., by the Dutch National Institute for Public Health and the Environment (RIVM). Such revisions may be helpful to ensure methods stay up-to-date.

It has been argued that to reduce the dependence on specific assumptions, different estimates could be combined into a consensus $R_t$ value or range. While in the United Kingdom meta-analysis techniques have been applied to this end [42], this is not without pitfalls. Unlike in classical meta-analysis, different estimates are typically obtained from the same data and thus inherently dependent. As pointed out by [43], this leads to estimators with unclear statistical properties. Moreover, when merging estimates based on different assumptions, the estimand becomes unclear, as do the assumptions underlying the consensus estimate. To combine estimates of the basic reproductive number $R_0$, an appealing approach where information is pooled separately for the generation time distribution and the epidemic growth rate has been suggested by [44]. This could likely be translated to $R_t$ estimation.

In the present work, we focused exclusively on estimates based on national-level case incidence data. We did not take into account regional or age stratification, which can be

incorporated e.g., in compartmental epidemiological models to estimate $R_t$ [4]. Reproductive numbers can also be estimated from other data streams including hospitalizations, deaths [16, 34], wastewater surveillance [37] and PCR cycle threshold data [45]. While these may resolve some of the issues of case incidences, e.g., their sensitivity to testing strategies, the dependence of estimates on analytical choices remains largely the same. Nonetheless, considering estimates based on various data streams may yield a more comprehensive picture. More generally, we underscore that the $R_t$ value should not be interpreted in isolation, but in conjunction with other epidemiological indicators like the overall case and hospitalization numbers or genetic data on the prevalence of different variants.

## Supporting information

**S1 Text. Supplementary text, figures and tables.** References to relevant code repositories and sources underlying Fig 2, additional remarks on the HZI approach, details on the handling of temporal shifts, additional figures on temporal coherence, extension of the Cori method to a conditional negative binomial distribution.
(PDF)

## Acknowledgments

The authors would like to thank Simas Kucinskas (globalrt) for making estimates available and providing additional information.

## Author Contributions

**Conceptualization:** Elisabeth K. Brockhaus, Johannes Bracher.

**Data curation:** Elisabeth K. Brockhaus, Daniel Wolffram, Jonas M. Littek, Anna J. Klesen, Laura M. Helleckes, Johannes Bracher.

**Formal analysis:** Elisabeth K. Brockhaus, Michael Osthege.

**Investigation:** Elisabeth K. Brockhaus, Johannes Bracher.

**Methodology:** Elisabeth K. Brockhaus, Sebastian Funk, Sam Abbott, Johannes Bracher.

**Project administration:** Johannes Bracher.

**Software:** Elisabeth K. Brockhaus, Daniel Wolffram, Michael Osthege, Laura M. Helleckes.

**Supervision:** Johannes Bracher.

**Validation:** Tanja Stadler, Michael Osthege, Tanmay Mitra, Ekaterina Krymova, Jana S. Huisman, Stefan Heyder, Laura M. Helleckes, Matthias an der Heiden, Sebastian Funk, Sam Abbott, Johannes Bracher.

**Visualization:** Elisabeth K. Brockhaus.

**Writing – original draft:** Elisabeth K. Brockhaus, Johannes Bracher.

**Writing – review & editing:** Elisabeth K. Brockhaus, Daniel Wolffram, Tanja Stadler, Michael Osthege, Tanmay Mitra, Jonas M. Littek, Ekaterina Krymova, Anna J. Klesen, Jana S. Huisman, Stefan Heyder, Laura M. Helleckes, Matthias an der Heiden, Sebastian Funk, Sam Abbott, Johannes Bracher.

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
