## [Decision Letter · Decision Letter 0]

29 Jun 2023

Dear Mr Bracher,

Thank you very much for submitting your manuscript "Why are different estimates of the effective reproductive number so different? A case study on COVID-19 in Germany" for consideration at PLOS Computational Biology.

As with all papers reviewed by the journal, your manuscript was reviewed by members of the editorial board and by several independent reviewers. In light of the reviews (below this email), we would like to invite the resubmission of a significantly-revised version that takes into account the reviewers' comments.

The Authors are expected to address all the criticisms by all Reviewers. In particular, include the results for HZI in Figure 5A & B (Reviewer #1), further elaborate the qualitative difference in Rt (e.g. Rt < or > 1) between different methods, provide (if possible) some insights on under what situations or methodological characteristics that would lead to most reliable estimates (Reviewer #2), further consider the generalizability of the study findings (Reviewer #3), such as when there is a much higher variability in Rt for Omicron or other situations, while it is understood that the study didn’t adopt a simulation approach which may give an advantage to methods most consistent with the data generating model. In additional to the above comments, please address,

1. Table 2, please avoid abbreviations, or provide the full terms in the footnote

We cannot make any decision about publication until we have seen the revised manuscript and your response to the reviewers' comments. Your revised manuscript is also likely to be sent to reviewers for further evaluation.

Sincerely,

Eric HY Lau, Ph.D.

Academic Editor

PLOS Computational Biology

Virginia Pitzer

Section Editor

PLOS Computational Biology

The Authors are expected to address all the criticisms by all Reviewers. In particular, include the results for HZI in Figure 5A & B (Reviewer #1), further elaborate the qualitative difference in Rt (e.g. Rt < or > 1) between different methods, provide (if possible) some insights on under what situations or methodological characteristics that would lead to most reliable estimates (Reviewer #2), further consider the generalizability of the study findings (Reviewer #3), such as when there is a much higher variability in Rt for Omicron or other situations, while it is understood that the study didn’t adopt a simulation approach which may give an advantage to methods most consistent with the data generating model. In additional to the above comments, please address,

1. Table 2, please avoid abbreviations, or provide the full terms in the footnote

Reviewer's Responses to Questions

**Comments to the Authors:**

Reviewer #1: I think the paper can be interesting to read and it can be helpful for understanding that different models with different assumptions can provide quite different results. Then we may wonder on how to align them and look if they are really incorrect or correct. In my opinion, the analysis was well-conceived and has logic. I recommend it for publication in PLoS CompBio. Some of my remarks, which are not critical, are below. However, my main concern is about keeping the analysis reliable in the "far" future. I am very unsure that many of those github repos listed in the Appendix would stay alive in two or in five years. So it would nice somehow to preserve links to stay working. Probably, by forking the original repos?

- the name for Section 2: I wonder why the authors called it "the agony of choice"? In my opinion, it is just about estimating the Rt by different methods. For example, when the meta-analysis performed, people do not usually give their personal assessment for analyzed studies. Here, when the authors called comparing all other studies as being in agony, they unintentionally become subjective.

- L197: the authors assess the importance of the generation time on estimation of Rt and risk assessment (L190-196), but write nothing about the incubation period. Why exactly do we need IP? Why changing the IP shifts Rt? (L199) This is not clear from the text.

- L139-L155 and further: could the authors indicate what all those acronyms mean? I understood RKI, but what's about ETH, SDSC, Ilmenau, etc.?

- Figure 5, panel AB: I wonder why I can't see HZI in the plots? If the width of 95% CI is zero, then there should be a line at zero, no?

- L339: I think it is "Cori et al." method.

- L403: Could the authors explain why they write "typically" lags? Does it mean that sometimes it may not be lagged by one generation time period? Could they expain more carefully the lagging issue?

Reviewer #2: The review is uploaded as an attachment.

Reviewer #3: In this study, the authors investigated why different estimates of estimates of the effective reproductive number are so different. The article presents an interesting perspective, but there are still some issues that need to be addressed urgently.

1. The authors used Germany as an example and found that many parameters will ultimately affect the estimation results. However, using only one example seems somewhat insufficient. We are very curious as to whether these results can be validated using simple and controllable simulation data. Modifying parameters (data source, data pre-processing, assumed generation time distribution, statistical tuning parameters and various delay distributions) to generate simulated data and then using these methods for estimation can help evaluate within-method temporal coherence and between-method agreement of retrospective estimates.

2. Page 5. The authors should summarize these methods in a table, listing the requirements for input data, the form of output results, and the calculation principles, etc.

3. The authors should organize the results in a clear and concise manner for better understanding, rather than presenting them in a large block of text, such as section 4.2.

4. The authors should explain what is meant by a "consolidated point" and further elaborate on the important role of the four indicators.

5. Figures 6 and 7 are difficult to understand. Besides the consistency among the methods, what does the mean absolute differences between point estimates represent?

**Have the authors made all data and (if applicable) computational code underlying the findings in their manuscript fully available?**

Reviewer #1: Yes

Reviewer #2: None

Reviewer #3: None

PLOS authors have the option to publish the peer review history of their article (what does this mean?). If published, this will include your full peer review and any attached files.

Reviewer #1: No

Reviewer #2: No

Reviewer #3: No
---

## [Decision Letter · Decision Letter 1]

3 Nov 2023

Dear Mr Bracher,

We are pleased to inform you that your manuscript 'Why are different estimates of the effective reproductive number so different? A case study on COVID-19 in Germany' has been provisionally accepted for publication in PLOS Computational Biology.

Best regards,

Eric HY Lau, Ph.D.

Academic Editor

PLOS Computational Biology

Virginia Pitzer

Section Editor

PLOS Computational Biology

Thanks for addressing all the editor’s and reviewers' comments. Congratulations on the excellent work!

Reviewer's Responses to Questions

**Comments to the Authors:**

Reviewer #2: I appreciate the authors' efforts in addressing my comments. I am satisfied with this revised version of the manuscript.

Reviewer #3: The revised version of the manuscript has been improved and I recommend it for publication.

**Have the authors made all data and (if applicable) computational code underlying the findings in their manuscript fully available?**

Reviewer #2: None

Reviewer #3: None

PLOS authors have the option to publish the peer review history of their article (what does this mean?). If published, this will include your full peer review and any attached files.

Reviewer #2: No

Reviewer #3: No

---

## [Editor Report · Acceptance letter]

21 Nov 2023

PCOMPBIOL-D-23-00673R1 

Why are different estimates of the effective reproductive number so different? A case study on COVID-19 in Germany

Dear Dr Bracher,

I am pleased to inform you that your manuscript has been formally accepted for publication in PLOS Computational Biology. Your manuscript is now with our production department and you will be notified of the publication date in due course.

With kind regards,

Anita Estes
